# Behavioural activation to prevent depression and loneliness among socially isolated older people with long-term conditions: The BASIL COVID-19 pilot randomised controlled trial

Simon Gilbody[1,2]*, Elizabeth Littlewood[1], Dean McMillan[1,2], Carolyn A. Chew-Graham[3], Della Bailey[1], Samantha Gascoyne[1], Claire Sloan[1], Lauren Burke[1], Peter Coventry[1], Suzanne Crosland[1], Caroline Fairhurst[1], Andrew Henry[1,4], Catherine Hewitt[1], Kalpita Joshi[1], Eloise Ryde[1,4], Leanne Shearsmith[5], Gemma Traviss-Turner[5], Rebecca Woodhouse[1], Andrew Clegg[5], Tom Gentry[6], Andrew J. Hill[5], Karina Lovell[7], Sarah Dexter Smith[4], Judith Webster[8], David Ekers[1,4]

1 Department of Health Sciences, University of York, York, United Kingdom, 2 Hull York Medical School, University of York, York, United Kingdom, 3 School of Medicine, Keele University, Staffordshire, United Kingdom, 4 Tees, Esk and Wear Valleys NHS FT, Research & Development, Flatts Lane Centre, Middlesbrough, United Kingdom, 5 Leeds Institute of Health Sciences, University of Leeds, Leeds, United Kingdom, 6 Age UK, London, United Kingdom, 7 Division of Nursing, Midwifery & Social Work, University of Manchester, Manchester, United Kingdom, 8 Patient and Public Representative, United Kingdom

* simon.gilbody@york.ac.uk

## Abstract

### Background

Older adults, including those with long-term conditions (LTCs), are vulnerable to social isolation. They are likely to have become more socially isolated during the Coronavirus Disease 2019 (COVID-19) pandemic, often due to advice to "shield" to protect them from infection. This places them at particular risk of depression and loneliness. There is a need for brief scalable psychosocial interventions to mitigate the psychological impacts of social isolation. Behavioural activation (BA) is a credible candidate intervention, but a trial is needed.

### Methods and findings

We undertook an external pilot parallel randomised trial (ISRCTN94091479) designed to test recruitment, retention and engagement with, and the acceptability and preliminary effects of the intervention. Participants aged ≥65 years with 2 or more LTCs were recruited in primary care and randomised by computer and with concealed allocation between June and October 2020. BA was offered to intervention participants (*n* = 47), and control participants received usual primary care (*n* = 49). Assessment of outcome was made blind to treatment allocation. The primary outcome was depression severity (measured using the Patient Health Questionnaire 9 (PHQ-9)). We also measured health-related quality of life (measured by the Short Form (SF)-12v2 mental component scale (MCS) and physical component scale (PCS)), anxiety (measured by the Generalised Anxiety Disorder 7 (GAD-7)), perceived social and emotional loneliness (measured by the De Jong Gierveld Scale: 11-item

**Data Availability Statement:** All relevant data are within the manuscript and its Supporting Information files.

**Funding:** SG, EL, DM, CCG, DB, PC, CH, GTT, AC, AH, KL and DE were funded by National Institute for Health Research (NIHR) Programme Grants for Applied Research (PGfAR) RP-PG-0217-20006 The funder had no role in study design, data collection and analysis, decision to publish, or preparation of the manuscript. The funder had no role in study design, data collection and analysis, decision to publish, or preparation of the manuscript.

**Competing interests:** We have read the journal's policy and the authors of this manuscript have the following competing interests: DE and CCG are current committee members for the NICE Depression Guideline (update) Development Group, and SG was a member between 2015-18. SG, PC and DMcM are supported by the NIHR Yorkshire and Humberside Applied Research Collaboration (ARC) and DE is supported by the North East and North Cumbria ARCs.

**Abbreviations:** AMD, adjusted mean difference; BA, behavioural activation; BASIL, Behavioural Activation in Social Isolation; BSW, BASIL support worker; CASPER, CollAborative care and active surveillance for Screen-Positive EldeRs with subthreshold depression; CI, confidence interval; CONSORT, Consolidated Standards of Reporting Trials; COPD, chronic obstructive pulmonary disease; COVID-19, Coronavirus Disease 2019; DASS, Depression Anxiety Stress Scales; GAD-7, Generalised Anxiety Disorder 7; GP, general practitioner; ICC, intercluster correlation; LTC, long-term condition; MCS, mental component scale; NHS, National Health Service; NIHR, National Institute for Health Research; PCS, physical component scale; PGfAR, Programme Grants for Applied Research; PHE, Public Health England; PHQ-9, Patient Health Questionnaire 9; PPI, patient and public involvement; PPI AG, patient and public involvement advisory group; QOF, Quality and Outcomes Framework; RCT, randomised controlled trial; SAP, Statistical Analysis Plan; SARS-CoV-2, Severe Acute Respiratory Syndrome Coronavirus 2; SD, standard deviation; SF, Short Form; TFA, Theoretical Framework of Acceptability; TIA, transient ischemic attack.

loneliness scale). Outcome was measured at 1 and 3 months. The mean age of participants was aged 74 years (standard deviation (SD) 5.5) and they were mostly White ($n = 92$, 95.8%), and approximately two-thirds of the sample were female ($n = 59$, 61.5%). Remote recruitment was possible, and 45/47 (95.7%) randomised to the intervention completed 1 or more sessions (median 6 sessions) out of 8. A total of 90 (93.8%) completed the 1-month follow-up, and 86 (89.6%) completed the 3-month follow-up, with similar rates for control (1 month: 45/49 and 3 months 44/49) and intervention (1 month: 45/47 and 3 months: 42/47) follow-up. Between-group comparisons were made using a confidence interval (CI) approach, and by adjusting for the covariate of interest at baseline. At 1 month (the primary clinical outcome point), the median number of completed sessions for people receiving the BA intervention was 3, and almost all participants were still receiving the BA intervention. The between-group comparison for the primary clinical outcome at 1 month was an adjusted between-group mean difference of −0.50 PHQ-9 points (95% CI −2.01 to 1.01), but only a small number of participants had completed the intervention at this point. At 3 months, the PHQ-9 adjusted mean difference (AMD) was 0.19 (95% CI −1.36 to 1.75). When we examined loneliness, the adjusted between-group difference in the De Jong Gierveld Loneliness Scale at 1 month was 0.28 (95% CI −0.51 to 1.06) and at 3 months −0.87 (95% CI −1.56 to −0.18), suggesting evidence of benefit of the intervention at this time point. For anxiety, the GAD adjusted between-group difference at 1 month was 0.20 (−1.33, 1.73) and at 3 months 0.31 (−1.08, 1.70). For the SF-12 (physical component score), the adjusted between-group difference at 1 month was 0.34 (−4.17, 4.85) and at 3 months 0.11 (−4.46, 4.67). For the SF-12 (mental component score), the adjusted between-group difference at 1 month was 1.91 (−2.64, 5.15) and at 3 months 1.26 (−2.64, 5.15). Participants who withdrew had minimal depressive symptoms at entry. There were no adverse events. The **B**ehavioural **A**ctivation in **S**ocial **I**solation (BASIL) study had 2 main limitations. First, we found that the intervention was still being delivered at the prespecified primary outcome point, and this fed into the design of the main trial where a primary outcome of 3 months is now collected. Second, this was a pilot trial and was not designed to test between-group differences with high levels of statistical power. Type 2 errors are likely to have occurred, and a larger trial is now underway to test for robust effects and replicate signals of effectiveness in important secondary outcomes such as loneliness.

## Conclusions

In this study, we observed that BA is a credible intervention to mitigate the psychological impacts of COVID-19 isolation for older adults. We demonstrated that it is feasible to undertake a trial of BA. The intervention can be delivered remotely and at scale, but should be reserved for older adults with evidence of depressive symptoms. The significant reduction in loneliness is unlikely to be a chance finding, and replication will be explored in a fully powered randomised controlled trial (RCT).

## Trial registration

ISRCTN94091479.

## Author summary

### Why was this study done?

- Older people with long-term conditions (LTCs) have been impacted by the Coronavirus Disease 2019 (COVID-19) pandemic and its restrictions. They are at risk of social isolation and, in turn, this could cause depression and loneliness, which are bad for health. Psychological approaches, such as behavioural activation (BA), could be helpful.

- A fair test is needed to demonstrate if BA can prevent the onset of depression and loneliness, but before we can do this, it is important to test this out in a smaller scale study.

### What did the researchers do and find?

- We designed a brief telephone-delivered intervention based on sound psychological principles known as BA. Here, we present the result of a pilot trial.

- We demonstrate that the intervention is acceptable to older people who are socially isolated as a consequence of the pandemic. We tested whether it is possible to collect important outcomes in the short term.

- There was some preliminary evidence that levels of loneliness were reduced at 3 months when BA is offered.

### What do these findings mean?

- This was a smaller scale pilot study, and our procedures worked well.

- If BA is shown to work, then this will be useful for policymakers in offering support to people who are socially isolated.

- This will also be useful once the COVID-19 pandemic has passed, since loneliness is common in older populations, and effective scalable solutions will be needed even after COVID-19.

- The **B**ehavioural **A**ctivation in **S**ocial **I**solation (BASIL) pilot trial was not designed to test differences in outcomes between the 2 groups. We will now test this in a much larger study.

## Introduction

In March 2020, a pandemic due to a new virus, the Severe Acute Respiratory Syndrome Coronavirus 2 (SARS-CoV-2), was declared. The first wave reached the United Kingdom within a short period of time, and, in March 2020, the UK governments (including devolved nations) administered a national stay-at-home order ("lockdown"), which included instructions for people to follow physical social distancing and self-isolation guidelines and recommendations

for strict isolation ("shielding") for the most vulnerable (such as those with long-term conditions (LTCs) and older people) in order to protect their own and others' health and to avoid a sudden increase in demand on the National Health Service (NHS). Shielding orders were eased in the second half of 2020 but were reintroduced in January 2021 as part of a further lockdown in response to subsequent Coronavirus Disease 2019 (COVID-19) waves. Many people with LTCs have remained avoidant of social contact in order to protect themselves from COVID-19 throughout the pandemic, irrespective of official guidance [1]. Similar recommendations and restrictions were also set in place in many healthcare systems around the world.

The mental health of the population has deteriorated during COVID-19 [2]. Many report social isolation, and the incidence of depression and anxiety has increased for older people and those with medical vulnerabilities [3]. A plausible mechanism for this is that COVID-19 restrictions have led to disruption of daily routine, loss of social contact and heightened isolation, and increased loneliness, which are each powerful precipitants of mental ill health [4]. Anticipating these behavioural and psychological consequences, a rapid review published in *The Lancet* [5] highlighted the detrimental impacts on mental health of quarantine, but offered limited advice on how this could be mitigated.

Social isolation, social disconnectedness, perceived isolation, and loneliness are known to be linked to common mental health problems, such as depression in older people [4,6]. The impairments in quality of life associated with depression are comparable to those of major physical illness [7]. Loneliness is a risk factor for depression and is also known to be detrimental to physical health and life expectancy [8,9].

Loneliness is not an inevitable consequence of social isolation, and strategies to prevent or mitigate loneliness were recognised as a population priority even before COVID-19 [10,11]. There are a number of promising interventions that focus on using social networks [12] or adapting the strategies central to cognitive behavioural therapy [13]. It is recognised that strategies that, for instance, maintain social connectedness could be important in ensuring the population mental health of older people [14], particularly during the pandemic [4] and in the planning for post-pandemic recovery [15]. If a brief effective intervention for depression and loneliness could be delivered at distance (such as via telephone) and at scale, then this would lead to significant benefits to the NHS and society. This could potentially mitigate the immediate and longer lasting psychological impacts of COVID-19 on vulnerable populations, including older people and those with LTCs [16].

Our research collaborative has previously developed, with input from older adults and carers, a credible intervention that can potentially meet these needs in populations of older people [17], and we have evaluated this in older populations with high rates of multiple LTCs [18,19]. Behavioural activation (BA) is a practical treatment that explores how physical inactivity and low mood are linked and result in a reduction of valued activity [20]. Within BA, the therapist and patient work together to develop a collaborative treatment plan that seeks to reinstate (or replace, if former activities are no longer possible) behaviours that connect people to sources of positive reinforcement (meaningful activity), including social connectedness. However, this has not yet been tested in a large-scale clinical trial or in the context of the COVID-19 pandemic where social isolation is more prevalent. Small-scale trials of BA delivered to socially isolated older people have produced encouraging preliminary results [21], but there is not yet sufficient research evidence to support whole-scale adoption or to inform the population response to COVID-19.

Along with many researchers working in the field of mental health, we were keen to use our existing research expertise and research capacity to help mitigate the impact of the COVID-19 pandemic. We therefore adapted our previous and ongoing work programme in early 2020 to

answer the following question: "Can we prevent or ameliorate depression and loneliness in older people with long-term conditions during isolation?".

In this paper, we present the rationale and results of a pilot randomised controlled trial (RCT) of manualised BA, adapted specifically to be delivered at scale and remotely (via the telephone or video call) for older adults who may have become socially isolated as a consequence of COVID-19. To the best of our knowledge, this is the first study to use behavioural approaches to prevent loneliness and depression in the COVID-19 pandemic. In this pilot RCT, we sought to assess the feasibility of recruiting and randomising participants to a trial of BA, of delivering the intervention, and of retaining participants in the trial. The primary clinical outcome was depression severity, as measured by the Patient Health Questionnaire 9 (PHQ-9), at 1 month post-randomisation; secondary clinical outcomes were health-related quality of life, anxiety, and perceived social and emotional loneliness.

## Methods

### Study design and participants

The **B**ehavioural **A**ctivation in **S**ocial **I**solation (BASIL) study is an external pilot RCT [22] and includes a concurrent qualitative study. The BASIL pilot is designed to provide key information on methods of recruitment, intervention uptake, retention, experience of the BA intervention for our target population, and acceptability of the intervention and training for intervention practitioners (hereafter BASIL support workers (BSWs)).

The COVID-19 responsive BASIL trials programme is supported by the National Institute for Health Research (NIHR) under grant RP-PG-0217-20006 and was adopted by the NIHR Urgent Public Health programme on May 28, 2020 [23]. The protocol for the BASIL pilot study was preregistered (ISRCTN94091479) on June 9, 2020, and recruitment took place between June 23 and October 15, 2020 (18 weeks in total). Older adults at risk of loneliness and depression as a consequence of social isolation under COVID-19 restrictions were recruited from primary care registers in the North East of England. They were randomised to receive either usual primary care from their general practice or BA intervention in addition to usual care (see below for full description of usual care and BA intervention).

- **Inclusion criteria:** Older adults (65 years or over) with 2 or more physical LTCs. The pragmatic definition and type of LTCs mirror that applied in primary care in the UK [24], and we focused on common LTCs experienced by older people (asthma/chronic obstructive pulmonary disease (COPD), diabetes, hypertension/coronary heart disease, and stroke) according to the primary care Quality and Outcomes Framework (QOF) [25], but also included conditions such as musculoskeletal problems and chronic pain. Participants included those subject to Government guidelines regarding COVID-19 self-isolation, social distancing, and shielding as relevant to their health conditions and age (although this was not a requirement and these requirements changed during the study period).

- **Exclusion criteria:** Older adults who have cognitive impairment, bipolar disorder/psychosis/psychotic symptoms, alcohol or drug dependence, in the palliative phase of illness, have active suicidal ideation, are currently receiving psychological therapy, or are unable to speak or understand English.

Potentially eligible patients were contacted by telephone by staff working with the general practices. Those patients who expressed an interest in the study during this initial telephone contact provided their verbal "permission to contact" for a member of the study team to

contact them by telephone to discuss the study and determine eligibility. Interested patients could also complete an online consent form or contact the study team directly.

## Randomisation, concealment of allocation, and masking

After consent, eligible participants completed a baseline questionnaire over the telephone with a study researcher. Consent was obtained, either in written form or via verbal consent [recorded] over the telephone. Participants were then randomised and informed of their group allocation (intervention or usual care with signposting). Participants were allocated in a 1:1 ratio using simple randomisation without stratification. Treatment allocation was concealed from study researchers at the point of recruitment using an automated computer data entry system, administered remotely by the York Trials Unit and using a computer-generated code. Owing to the nature of the intervention, none of the participants, general practices, study clinicians, or BSWs could be blinded to treatment allocation. General practitioners (GPs) were informed by letter of participant treatment allocation. Outcome assessment was by self-report, and study researchers facilitating the telephone-based outcome assessment were blind to treatment allocation.

## Intervention (BA)

The intervention (BA within a collaborative care framework) has been described elsewhere [18] and was adapted for the purposes of the BASIL trial. Within the BASIL BA intervention, the therapist (BSW) and participant worked together to develop a collaborative treatment plan that sought to reinstate (or replace, if former activities were no longer possible because of social isolation and/or LTCs) behaviours that connect them to sources of positive reinforcement (valued activity). BA has the potential to address depression and loneliness in the presence of social isolation in this way [17,26] and the simplicity of BA made it suitable for delivery in the context of COVID-19.

Intervention participants were offered up to 8 sessions over a 4- to 6-week period delivered by trained BSWs, accompanied by participant materials. Participants in the intervention group were provided with a BASIL BA workbook. This booklet was modified to take account of government guidance regarding the need for social isolation/physical distancing and enforced isolation for those people most at risk ("extremely vulnerable" people). For example, the BASIL booklet discussed ways to replace activities that are no longer possible with ones that preserve social distancing while helping participants stay connected with the activities and people important to them; illustrative patient stories included in the booklet were modified to take account of COVID-19 restrictions. Examples of replacement activities from the BASIL self-help manual are presented in Box 1. BA acknowledged the disruption to people's lives and usual routines and encouraged the establishment of a balanced daily routine. The intervention also recognised that participants may be worried about the current situation due to COVID-19 and suggested strategies to help cope.

All intervention sessions were delivered remotely via telephone or video call, according to participant preference. The first session was scheduled to last approximately 1 hour, with subsequent sessions lasting approximately 30 minutes.

Depression symptom monitoring at each intervention session was undertaken using a validated depression scale (the Depression Anxiety Stress Scales (DASS) [27]) with scores guiding decision-making by BSWs and guided by supervision provided by clinical members of the study team. Where risk or significant clinical deterioration was noted, the participant was supported to access more formal healthcare interventions (including mental health care) via their GP. Where feasible and where considered appropriate and acceptable by the participant and

> ### Box 1. BASIL examples of replacement activities using functional equivalence. See [17] for a description of the principles of BA and its application in older people with LTCs
>
> BA pays particular attention to the function the behaviour holds for an individual and that reinforcement is determined functionally. An important consequence of this view is the idea of functional equivalence. A specific form of a behaviour may have served a particular function for a person; however, that behaviour may no longer be possible due to physical health problems or COVID-19 lockdown. In this situation, an aim of treatment was to identify a functionally equivalent behaviour that is different and therefore still possible despite physical changes or shielding, but which may serve the same function for a person. Below are 2 illustrative examples of functionally equivalent goals from the BASIL self-help manual.
>
> #### Functionally equivalent goal, example 1
>
> Sandra decided that she would like to increase her activity levels and said she would have liked to go to the Age UK exercise group. However, Sandra realised that this will not be happening due to the need for physical distancing. The BASIL Support Worker advised that there are exercises online, and she could help Sandra access these. In addition, many TV channels are showing exercises that can be done at home.
>
> #### Functionally equivalent goal, example 2
>
> Sandra would like to have a cup of tea with her friend Shirley but realises that this is not possible. She and Shirley used to meet up on a regular basis, before the ministroke (TIA), and she really missed their chats. She thought that she might suggest a regular 'virtual meeting', rather than just a quick telephone call, if the Support Worker can help her gain confidence with using her smart phone.

BSW, the intervention was extended to include involvement of a participant's informal caregiver/significant other. Intervention participants continued to receive their usual care/treatment (where this was feasible given COVID-19) alongside the BASIL intervention, and no treatment was withheld.

## Comparator (usual GP care)

Participants in the control group received usual care as provided by their current NHS and/or third sector providers. In addition, control participants were "signposted" to reputable sources of self-help and information, including advice on how to keep mentally and physically well. Examples of such sources was the Public Health England's (PHE) "Guidance for the public on the mental health and wellbeing aspects of coronavirus (COVID-19)" [28] and Age UK [29].

## Outcome measures

Demographic information was obtained at baseline and included age, sex, LTC type, socioeconomic status, ethnicity, education, marital status, and number of children.

The primary objective for this pilot trial was to obtain estimates of key feasibility measures including rates of recruitment, randomisation, retention, intervention delivery, and engagement.

The primary clinical outcome was self-reported symptoms of depression, assessed by the PHQ-9 [30]. The PHQ-9 was also administered at screening to ascertain risk of self-harm or suicide and again at baseline, 1 and 3 months post-randomisation. The primary time point was 1 month. Other secondary clinical outcomes measured at baseline, 1 and 3 months, were health-related quality of life (measured by the Short Form (SF)-12v2 mental component scale (MCS) and physical component scale (PCS)) [31], anxiety (measured by the GAD-7) [32], perceived social and emotional loneliness (measured by the De Jong Gierveld Scale: 11-item loneliness scale), and questions relating to COVID-19 circumstances and adherence to government guidelines [33]. The PHQ-9 is scored from 0 to 27, the GAD-7 from 0 to 21, and the De Jong Gierveld Loneliness Scale from 0 to 11 (emotional subscale from 0 to 6 and social from 0 to 5), for each a higher score indicates a worse outcome. The physical and mental health component scores of the SF-12v2 range from 0 to 100, where a higher score indicates better health. We also tested our ability to collect resource use data, but these data are not summarised or described, since there was no planned economic evaluation in this pilot trial.

## Sample size and statistical analysis

**Sample size.**   The primary aim of the BASIL pilot trial was to test the feasibility of the intervention and the methods of recruitment, randomisation, and follow-up [22]. Sample size calculations were based on estimating attrition and standard deviation (SD) of the primary outcome. We aimed to recruit 100 participants. The intervention was delivered by BSWs and allowed for potential clustering by BSWs assuming an intercluster correlation (ICC) of 0.01 and mean cluster size of 15 based upon previous studies [18]. The effective sample size was therefore 88. Anticipating 15% to 20% of participants would be lost to follow-up (based on 17% in the CollAborative care and active surveillance for Screen-Positive EldeRs with sub-threshold depression (CASPER) trial of older adults [18]), this would result in an effective sample size of at least 70 participants, which is sufficient to allow reasonably robust estimates of the SD of the primary outcome measure to inform the sample size calculation for a definitive trial [34,35]. See Statistical Analysis Plan (SAP) in S1 Data.

**Statistical analysis.**   This study is reported as per the Consolidated Standards of Reporting Trials (CONSORT) guideline (see S2 Data). The flow of participants through the pilot trial (number of people identified, approached, screened, eligible, randomised, receiving the intervention, and providing outcome data) is detailed in a CONSORT flow diagram as per pilot trial recommendations [Fig 1] [22,36]. The number of individuals withdrawing from the intervention and/or the trial, and any reasons for withdrawal, was summarised by trial arm. All baseline and outcome data were summarised descriptively, by trial arm, using mean and SD for continuous outcomes, and count and percentage for categorical data. To quantify the acceptability of the intervention, the number and duration of sessions were summarised.

For each of the clinical outcomes, missing item level data were handled according to the user guides. No other methods for imputing missing data were employed, which was deemed appropriate as there were minimal missing data, and this was a pilot trial where all analyses were purely exploratory. Linear regression was used to explore differences in the clinical outcomes, adjusting for the baseline measure of the score as a covariate (ANCOVA approach), between groups at 1 and 3 months. Model assumptions were assessed, and no concerning deviations or observations were observed. The adjusted mean difference (AMD) and 95% confidence interval (CI) was reported as preliminary estimates of effect, but this pilot trial was not powered to show efficacy. We attach an SAP as a Supporting information (S1 Data).

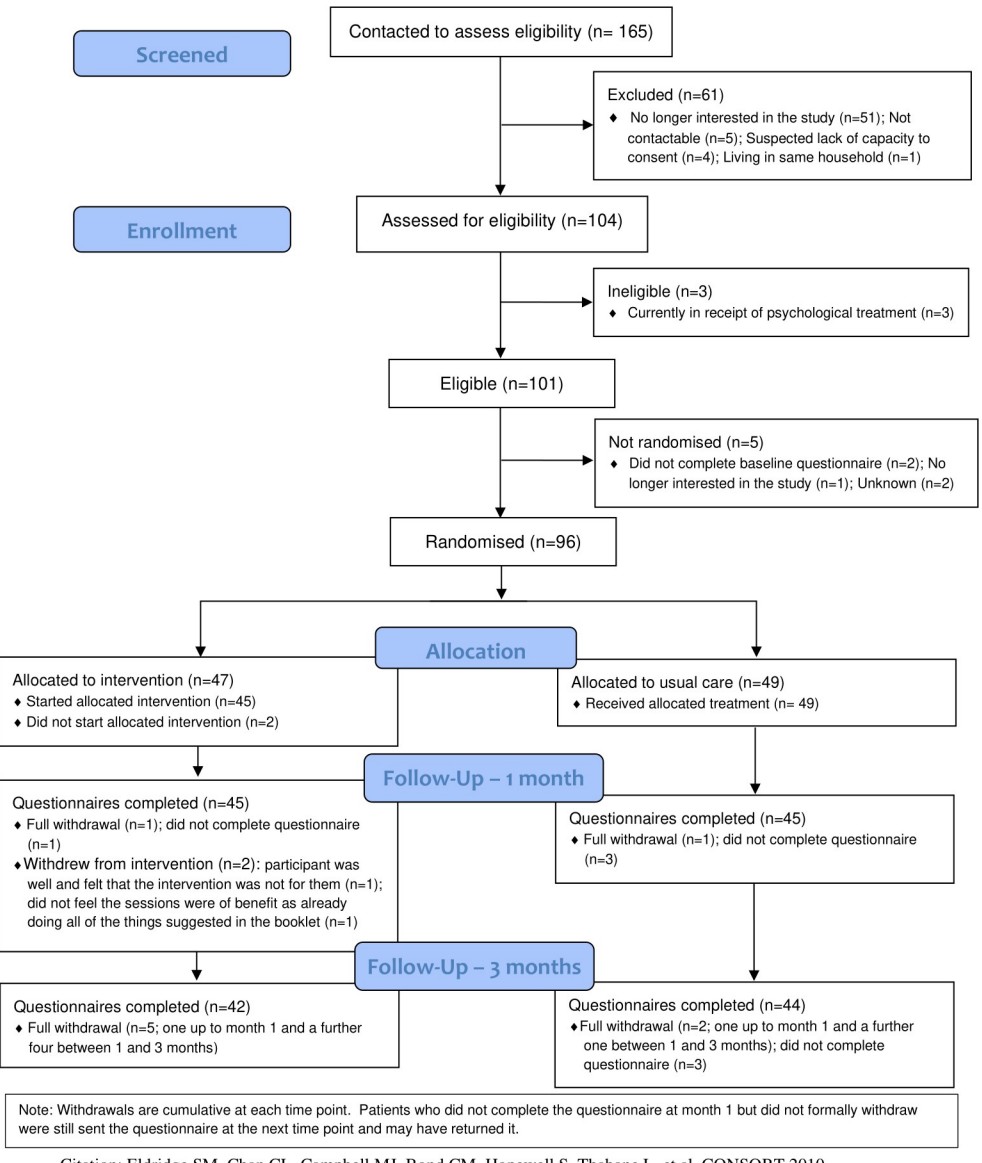

Note: Withdrawals are cumulative at each time point. Patients who did not complete the questionnaire at month 1 but did not formally withdraw were still sent the questionnaire at the next time point and may have returned it.

Citation: Eldridge SM, Chan CL, Campbell MJ, Bond CM, Hopewell S, Thabane L, et al. CONSORT 2010 statement: extension to randomised pilot and feasibility trials. BMJ. 2016;355.

**Fig 1. CONSORT flow diagram.**

## Process evaluation

A nested qualitative study was conducted to provide important learning about the study processes and acceptability of the BASIL intervention. We planned semi-structured interviews of up to 15 participants who completed the BASIL intervention ("completers"), up to 10 participants who did not complete the intervention ("non-completers"), and all BSWs who delivered the intervention (*n* = 9). Interviews explored views and experiences of the study and acceptability of the intervention. Initial thematic analysis [37] and subsequent analysis sensitised by the Theoretical Framework of Acceptability (TFA) [38] were undertaken.

### Patient and public involvement (PPI)

The BASIL trial was informed by a patient and public involvement advisory group (PPI AG) who were working with the research collective on the existing NIHR-funded research programme. This PPI AG included older adults with lived experience of mental health and/or physical health conditions and caregivers. The PPI AG were consulted on many aspects of the trial design including modification of the BA intervention for BASIL, remote recruitment of BASIL participants, and the relevance and readability of study recruitment information. The group are a vital component of the BASIL trials programme and will continue to contribute throughout the delivery of this work.

### Role of funding source

This project was funded by the NIHR Programme Grants for Applied Research (PGfAR) programme (RP-PG-0217-20006). The scope of our preexisting research into multimorbidity in older people was extended at the outset of the COVID-19 pandemic with the agreement of the funder to consider loneliness and depression in this vulnerable group. The NIHR PGfAR programme had no role in the writing of this manuscript or the decision to submit it for publication.

### Ethical approval

Ethical approval for the study was granted by Yorkshire and The Humber—Leeds West Research Ethics Committee on April 23, 2020, Yorkshire and The Humber—Leeds West Research Ethics Committee (The Old Chapel, Royal Standard Place, Nottingham, NG1 6FS, UK; +44 (0)207 104 8018; leedswest.rec@hra.nhs.uk), ref: 18/YH/0380 (approved as substantial amendment 02 under existing NIHR IRAS249030 research programme).

## Results

### Participant recruitment and characteristics

Database searches were conducted in 2 general practices within Tees, Esk and Wear Valleys NHS Foundation Trust. A total of 799 study information packs were mailed out across the 2 practices between June 17 and September 4, 2020 (initially in batches of 50, later increased to 100). Of these, 104 were screened for eligibility, and 96 were recruited out of our target of 100: 3 were not eligible as they were currently in receipt of psychological treatment, 2 eligible participants did not complete the baseline questionnaire following consent, 1 was no longer interested in the study, and for 2 more, the reason for nonparticipation was not provided. Participants were randomised between June 23 and October 15, 2020: 47 to the BA intervention group and 49 to usual care with signposting group (Fig 1).

The mean age of participants was aged 74 years (SD 5.5) and they were mostly White (*n* = 92, 95.8%), and approximately two-thirds of the sample were female (*n* = 59, 61.5%) (Table 1). Cardiovascular conditions (49.0%) and arthritis (38.5%) were the most commonly reported long-term health conditions. The majority of participants (55.2%) were social/physical distancing and adhering to UK Government's guidance in relation to COVID-19 restrictions all of the time (65.6%). There was reasonable balance in baseline characteristics between the 2 groups, but with some differences including a larger proportion of females, and current and former smokers, and fewer participants shielding in the usual care group than the intervention group; however, given the sample size, each participant per group equates to approximately 2 percentage points, so even small numerical imbalances equate to more noticeable imbalances in proportions.

Table 2 presents summaries of the clinical outcomes by time point and treatment group. Unadjusted between-group mean differences tended to favour the intervention.

**Table 1. Baseline demographics of participants as randomised.**

| Demographic | | Usual care (n = 49) | Intervention (n = 47) | Total (n = 96) |
|---|---|---|---|---|
| **Age, years** | Mean (SD) | 74.1 (5.6) | 74.2 (5.4) | 74.2 (5.5) |
| **Sex, n (%)** | Male | 17 (34.7) | 20 (42.6) | 37 (38.5) |
| | Female | 32 (65.3) | 27 (57.4) | 59 (61.5) |
| **Ethnicity, n (%)** | White | 47 (95.9) | 45 (95.7) | 92 (95.8) |
| | Black or Black British | 0 (0.0) | 0 (0.0) | 0 (0.0) |
| | Asian or Asian British | 0 (0.0) | 0 (0.0) | 0 (0.0) |
| | Other | 2 (4.1) | 2 (4.3) | 4 (4.2) |
| **[†]LTC type, n (%)** | Cardiovascular condition | 21 (42.9) | 26 (55.3) | 47 (49.0) |
| | Arthritis | 21 (42.9) | 16 (34.0) | 37 (38.5) |
| | Respiratory condition | 17 (34.7) | 18 (38.3) | 35 (36.5) |
| | Diabetes | 14 (28.6) | 14 (29.8) | 28 (29.2) |
| | Stroke | 5 (10.2) | 4 (8.5) | 9 (9.4) |
| | Chronic pain | 2 (4.1) | 3 (6.4) | 5 (5.2) |
| | Osteoporosis | 4 (8.2) | 2 (4.3) | 6 (6.3) |
| | Neurological condition | 1 (2.0) | 0 (0.0) | 1 (1.0) |
| | Cancer | 1 (2.0) | 1 (2.1) | 2 (2.1) |
| | Other | 27 (55.1) | 21 (44.7) | 48 (50.0) |
| **Smoking status, n (%)** | I have never smoked | 16 (32.7) | 22 (46.8) | 38 (39.6) |
| | I currently smoke | 5 (10.2) | 7 (14.9) | 12 (12.5) |
| | I am an ex-smoker | 28 (57.1) | 18 (38.3) | 46 (47.9) |
| **Alcohol intake** | Yes | 7 (14.3) | 6 (12.8) | 13 (13.5) |
| **(3+ units daily), n (%)** | No | 42 (85.7) | 41 (87.2) | 83 (86.5) |
| **Post-16 education, n (%)** | Yes | 29 (59.2) | 32 (68.1) | 61 (63.5) |
| **Degree or equivalent, n (%)** | Yes | 18 (36.7) | 19 (40.4) | 37 (38.5) |
| **Marital status, n (%)** | Single | 1 (2.0) | 0 (0.0) | 1 (1.0) |
| | Divorced/separated | 11 (22.4) | 9 (19.1) | 20 (20.8) |
| | Widowed | 11 (22.4) | 10 (21.3) | 21 (21.9) |
| | Cohabiting | 0 (0.0) | 1 (2.1) | 1 (1.0) |
| | Civil partnership | 0 (0.0) | 0 (0.0) | 0 (0.0) |
| | Married | 26 (53.1) | 27 (57.4) | 53 (55.2) |
| **Number of children, n (%)** | 0 | 3 (6.1) | 3 (6.4) | 6 (6.3) |
| | 1 | 7 (14.3) | 8 (17.0) | 15 (15.6) |
| | 2 | 24 (49.0) | 16 (34.0) | 40 (41.7) |
| | 3 | 10 (20.4) | 15 (31.9) | 25 (26.0) |
| | 4+ | 5 (10.2) | 5 (10.6) | 10 (10.4) |
| **How many people do you share your home with? n (%)** | Live alone | 22 (44.9) | 18 (38.3) | 40 (41.7) |
| | 1 person | 26 (53.1) | 25 (53.2) | 51 (53.1) |
| | 2 people | 1 (2.0) | 3 (6.4) | 4 (4.2) |
| | 3 people | 0 (0.0) | 1 (2.1) | 1 (1.0) |
| | 4 or more people | 0 (0.0) | 0 (0.0) | 0 (0.0) |
| **Current circumstance, n (%)** | Social/physical distancing | 29 (59.2) | 24 (51.1) | 53 (55.2) |
| | Self-isolating without COVID-19 symptoms | 10 (20.4) | 4 (8.5) | 14 (14.6) |
| | Self-isolating with COVID-19 symptoms | 0 (0.0) | 0 (0.0) | 0 (0.0) |
| | Shielding* | 10 (20.4) | 19 (40.4) | 29 (30.2) |
| | Other | 0 (0.0) | 0 (0.0) | 0 (0.0) |

(*Continued*)

**Table 1.** (Continued)

| Demographic | | Usual care (n = 49) | Intervention (n = 47) | Total (n = 96) |
|---|---|---|---|---|
| **Adherence to UK Government's guidance in relation to COVID-19 restrictions, n (%)** | All of the time | 31 (63.3) | 32 (68.1) | 63 (65.6) |
| | Most of the time | 15 (30.6) | 15 (31.9) | 30 (31.3) |
| | Some of the time | 3 (6.1) | 0 (0.0) | 3 (3.1) |
| | A little of the time | 0 (0.0) | 0 (0.0) | 0 (0.0) |
| | None of the time | 0 (0.0) | 0 (0.0) | 0 (0.0) |

† Conditions are not mutually exclusive so percentages not expected to sum to 100.

* Shielding was self-defined as following government guidance on shielding due to being a clinically highly vulnerable category. The definitions and list of categories changed according to shifting guidance during the course of the study.

Other conditions include (all listed for a single person unless otherwise indicated in brackets, but people may have listed more than one other condition): USUAL CARE—underactive thyroid/other thyroid problem (n = 5), high blood pressure (n = 4), high cholesterol (n = 2), sciatica (n = 2), depression (n = 2), non-chronic pain in in back and legs, "tablets for bones," autonomic neuropathic dysfunction (mainly affects bowels), brain tumour, Crohn disease, deaf in one ear from birth, flat feet, spondylitis in the spine, anxiety, kidney problems, previous stroke (n = 2), previous history of cancer, nerve problems in back, hernia in lumbar disc, Barrett disease, emphysema, sleep apnea, kidney infections/sepsis, anemia, digestive problems, pacemaker in stomach, angina, ileostomy from ulcerative colitis, glaucoma in one eye, right total hip replacement due to arthritis, raised cholesterol, some cardiovascular disease—short of breath on exertion, retinitis pigmentosa (eye sight), Reynaud syndrome, skin condition, lichen sclerosus, skin lupus; INTERVENTION—underactive thyroid or other thyroid issue (n = 7), high blood pressure (n = 3), depression (n = 3), previous cancer (n = 2), atrial fibrillation, angina (n = 2), premature ventricular contractions, emphysema, joint pain, lung function issues due to asbestos exposure, sciatica, mild bronchiectasis, eczema, IBS, Sturge–Weber syndrome, vertigo, tinnitus, chronic kidney disease, polymyalgia, high cholesterol, kidney cyst, and gall stones.

COVID-19, Coronavirus Disease 2019; IBS, irritable bowel syndrome; LTC, long-term condition; SD, standard deviation.

## Engagement with the BASIL intervention

Levels of engagement with the BA intervention were high. Of the 47 intervention participants randomised to the BA intervention group, 45 (95.7%) commenced the intervention, with 44 participants completing 2 or more sessions. The number of sessions completed range from 0 to 8 (median of 6 sessions) out of a total of up to 8 sessions. Participants preferred telephone over video contact. Sessions lasted a mean of 36.7 minutes (SD 15.7). Two participants withdrew from the intervention (after completing 1 and 2 sessions, respectively); one participant stated their reason for withdrawal was that they felt "well," and the intervention was "not for them" as they were already engaging in BA-related activities. At 1 month (the primary clinical outcome point), the median number of completed sessions for people receiving the BA intervention was 3, and almost all participants were still receiving the BA intervention.

## Retention, follow-up, withdrawal, and completeness of data

Of the 96 participants randomised into the study, 90 (93.8%) completed the 1-month follow-up, and 86 (89.6%) completed the 3-month follow-up. Reasons for withdrawal include personal reasons, family bereavement, and finding the study/study questions upsetting and anxiety provoking.

Data completeness was good with all patient-reported outcome measures (PHQ-9, GAD-7, De Jong Gierveld Loneliness Scale, and SF-12v2).

## Outcome data and between group comparisons at 1 and 3 months

The adjusted mean difference (AMD) between groups in the PHQ-9 indicated lower severity in the intervention group at 1 month (−0.50, 95% CI −2.01 to 1.01) and the usual care group at 3 months (0.19, 95% CI −1.36 to 1.75) (Table 3). In De Jong Gierveld score, the AMD indicated

**Table 2. Patient-reported outcome measures.**

| Outcome measure | Control | Intervention |
|---|---|---|
| **PHQ-9,** *n*, mean (SD) | | |
| Baseline | 49, 7.5 (6.2) | 47, 6 (5.6) |
| 1 month | 45, 6.3 (5.9) | 45, 4.9 (4.6) |
| 3 months | 44, 5.7 (5.3) | 42, 5.3 (5.4) |
| **PHQ-9 categories baseline**, *n* (%) | | |
| Minimal depression (0 to 4) | 22 (44.9) | 21 (44.7) |
| Mild depression (5 to 9) | 10 (20.4) | 14 (29.8) |
| Moderate depression (10 to 14) | 11 (22.4) | 9 (19.1) |
| Moderately severe depression (15 to 19) | 3 (6.1) | 1 (2.1) |
| Severe depression (20 to 27) | 3 (6.1) | 2 (4.3) |
| **PHQ-9 categories 1 month**, *n* (%) | | |
| Minimal depression (0 to 4) | 22 (48.9) | 24 (53.3) |
| Mild depression (5 to 9) | 12 (26.7) | 12 (26.7) |
| Moderate depression (10 to 14) | 6 (13.3) | 8 (17.8) |
| Moderately severe depression (15 to 19) | 3 (6.7) | 1 (2.2) |
| Severe depression (20 to 27) | 2 (4.4) | 0 (0.0) |
| **PHQ-9 categories 3 months**, *n* (%) | | |
| Minimal depression (0 to 4) | 21 (47.7) | 23 (54.8) |
| Mild depression (5 to 9) | 15 (34.1) | 11 (26.2) |
| Moderate depression (10 to 14) | 4 (9.1) | 4 (9.5) |
| Moderately severe depression (15 to 19) | 3 (6.8) | 4 (9.5) |
| Severe depression (20 to 27) | 1 (2.3) | 0 (0.0) |
| **GAD-7,** *n*, mean (SD) | | |
| Baseline | 49, 5.2 (5.8) | 47, 3.8 (4.8) |
| 1 month | 45, 4.2 (5.1) | 45, 3.6 (4.2) |
| 3 months | 44, 3.7 (5.0) | 42, 3.5 (3.9) |
| **GAD-7 categories baseline**, *n* (%) | | |
| Anxiety (0 to 4) | 31 (63.3) | 35 (74.5) |
| Mild anxiety (5 to 9) | 9 (18.4) | 7 (14.9) |
| Moderate anxiety (10 to 14) | 4 (8.2) | 3 (6.4) |
| Severe anxiety (15 to 21) | 5 (10.2) | 2 (4.3) |
| **GAD-7 categories 1 month**, *n* (%) | | |
| Anxiety (0 to 4) | 31 (68.9) | 30 (66.7) |
| Mild anxiety (5 to 9) | 8 (17.8) | 10 (22.2) |
| Moderate anxiety (10 to 14) | 3 (6.7) | 4 (8.9) |
| Severe anxiety (15 to 21) | 3 (6.7) | 1 (2.2) |
| **GAD-7 categories 3 months**, *n* (%) | | |
| Anxiety (0 to 4) | 31 (70.5) | 28 (66.7) |
| Mild anxiety (5 to 9) | 9 (20.5) | 9 (21.4) |
| Moderate anxiety (10 to 14) | 1 (2.3) | 4 (9.5) |
| Severe anxiety (15 to 21) | 3 (6.8) | 1 (2.4) |
| **De Jong Gierveld Loneliness Scale**, *n*, mean (SD) | | |
| Baseline | 49, 5.1 (3.2) | 47, 4.6 (3.5) |
| 1 month | 45, 4.6 (3.1) | 45, 4.7 (3.0) |
| 3 months | 44, 5.0 (3.0) | 42, 4.1 (2.9) |
| **De Jong Gierveld Emotional Loneliness Subscale**, *n*, mean (SD) | | |
| Baseline | 49, 3.1 (1.9) | 47, 3.0 (2.0) |

(*Continued*)

**Table 2.** (Continued)

| Outcome measure | Control | Intervention |
|---|---|---|
| 1 month | 45, 3.0 (1.7) | 45, 3.1 (1.9) |
| 3 months | 44, 3.4 (1.7) | 42, 3.0 (1.7) |
| De Jong Gierveld Social Loneliness Subscale, *n*, mean (SD) | | |
| Baseline | 49, 2.0 (1.8) | 47, 1.6 (1.8) |
| 1 month | 45, 1.5 (1.8 | 45, 1.6 (1.8) |
| 3 months | 44, 1.6 (1.8) | 42, 1.1 (1.6) |
| **SF-12v2 (physical component score)**, *n*, mean (SD) | | |
| Baseline | 49, 39.4 (10.7) | 47, 40.3 (11.3) |
| 1 month | 45, 40.0 (10.5) | 45, 41.4 (12.4) |
| 3 months | 44, 41.0 (11.5) | 42, 41.8 (11.7) |
| **SF-12v2 (mental component score)**, *n*, mean (SD) | | |
| Baseline | 49, 47.0 (13.9) | 47, 48.9 (10.5) |
| 1 month | 45, 48.4 (13.0) | 45, 52.0 (9.5) |
| 3 months | 44, 49.0 (11.5) | 42, 51.1 (9.7) |

Range of possible scores: PHQ-9 0 to 27; GAD-7 0 to 21; and De Jong Gierveld total 0 to 11—emotional subscale 0 to 6, social subscale 0 to 5 (higher indicates worse outcome). SF-12v2 subscales 0 to 100 (higher indicates better outcome).

GAD-7, Generalised Anxiety Disorder 7; PHQ-9, Patient Health Questionnaire 9; SD, standard deviation; SF, Short Form.

lower severity in the usual care group at 1 month (0.28, 95% CI −0.51 to 1.06) and the intervention group at 3 months (−0.87, 95% CI −1.56 to −0.18) when the 95% CI did not contain 0 (Table 3).

## Process evaluation and changes to the intervention in light of participant feedback

Intervention participants were invited to be interviewed following completion of the 1-month follow-up and conclusion of their participation in the BASIL pilot intervention. Semi-structured interviews were conducted with 15 participants who completed the BA intervention. A study participant who did not complete the BASIL intervention was interviewed as a "non-completer." All 9 BSWs who delivered the BASIL intervention were interviewed between August 2020 and November 2020. All interviews were conducted over the telephone, digitally recorded with consent and transcribed verbatim. Transcripts formed the basis for analysis. We summarise the key findings from a thematic analysis [37] and subsequent changes to the BASIL intervention ahead of the BASIL main trial. Analysis sensitised by the TFA [38] will be reported separately. Intervention participant and BSW demographics are reported in S1 and S2 Tables. Two researchers (CCG and CS) undertook the qualitative data analysis, coding transcripts, and reviewing/developing a coding frame.

**Recruitment and study eligibility.** Study recruitment methods appeared to be generally acceptable and clear:

> "The doctor gives you a warning that this is about to happen [be contacted] and then you're prepared when somebody phones up that it's not a scam, a con, which is what I don't worry about it because I know how to deal with it but to some people it could be worrying."
> (OA16)

**Table 3.  Unadjusted and AMD between groups at 1 and 3 months for primary and secondary outcomes measure.**

|  | Unadjusted mean difference (95% CI) | AMD[a] (95% CI) |
|---|---|---|
| *1 month* |  |  |
| PHQ-9 | −1.44 (−3.66, 0.77) | −0.50 (−2.01, 1.01) |
| GAD-7 | −0.54 (−2.52, 1.44) | 0.20 (−1.33, 1.73) |
| De Jong Gierveld Scale (total) | 0.13 (−1.14, 1.41) | 0.28 (−0.51, 1.06) |
| De Jong Gierveld Emotional Loneliness Subscale | 0.07 (−0.68, 0.81) | 0.14 (−0.39, 0.67) |
| De Jong Gierveld Social Loneliness Subscale | 0.07 (−0.68, 0.81) | 0.14 (−0.42, 0.69) |
| SF-12v2 (physical component score)[b] | 1.40 (−3.42, 6.22) | 0.34 (−4.17, 4.85) |
| SF-12v2 (mental component score)[b] | 3.60 (−1.17, 8.37) | 1.91 (−2.64, 5.15) |
| *3 months* |  |  |
| PHQ-9 | −0.39 (−2.70, 1.91) | 0.19 (−1.36, 1.75) |
| GAD-7 | −0.16 (−2.09, 1.78) | 0.31 (−1.08, 1.70) |
| De Jong Gierveld Scale (total) | −0.86 (−2.14, 0.43) | −0.87 (−1.56, −0.18) |
| De Jong Gierveld Emotional Loneliness Subscale | −0.36 (−1.09, 0.36) | −0.37 (−0.85, 0.11) |
| De Jong Gierveld Social Loneliness Subscale | −0.50 (−1.22, −0.23) | −0.50 (−1.00, −0.01) |
| SF-12v2 (physical component score)[b] | 0.81 (−4.16, 5.77) | 0.11 (−4.46, 4.67) |
| SF-12v2 (mental component score)[b] | 2.09 (−2.48, 6.65) | 1.26 (−2.64, 5.15) |

[a] Adjusted for baseline measure of the outcome as a covariate.

[b] For the SF12, a positive score indicates direction of effect in favour of in intervention group, and a negative score indicates direction of effect in favour of control group.

AMD, adjusted mean difference; CI, confidence interval; GAD-7, Generalised Anxiety Disorder 7; PHQ-9, Patient Health Questionnaire 9.

Some participants, generally participants without symptoms of depression at study entry and also some BSWs, raised the importance of more targeted recruitment to the BASIL intervention:

> "I do think just some consideration needs to be given to who we're targeting, maybe it's not quite so useful for people on the threshold of depression and feel that they're doing quite well." (BSW 04)

> "I think possibly it needs to be more targeted, so anybody who has a painful medical condition or who lives alone who is isolated. Certainly I think it would benefit [people who are isolated] a lot. I think the wide spread that you have currently can be more targeted. [it could be] more focused and more helpful." (OA02)

**Intervention delivery and content.**   Remote delivery of the intervention by telephone was acceptable. Although video calls were offered, these were not taken up by participants. Some participants and BSWs reported they would have preferred face-to-face intervention delivery, had this been possible. One participant suggested that those with hearing difficulties would find telephone delivery more difficult. The number and frequency of intervention sessions were acceptable, although one participant reported that they would prefer 1 session per week to allow them time to implement agreed activities and plans. Some BSWs reported that it could be difficult to stick to the 30-minute timing for more complex or isolated cases, where meetings took longer.

The BASIL BA self-help booklet was thought to be engaging, and people found the mood/behaviour cycle understandable. However, some participants—those with few depression

symptoms at study entry—found this model of limited relevance. Several participants reported that they would use the booklet after the intervention ended:

> "So, in days of darkness I'll be able to flick through it [the booklet] and say, that's what that was all about, how to break things down and not get upset about them and not let them get you down." (OA06)

The patient stories in the booklet were reported to be relevant, although some participants reported that booklet activity examples could ideally be more varied. Both BSWs and participants found activity planning to be sometimes difficult, especially where some services were shut under lockdown conditions. Planned activities may therefore need to be sufficiently flexible to accommodate changes in COVID-19 restrictions.

**Study adaptations.**   The process evaluation led to intervention adaptations for the BASIL main trial, including refining the study eligibility criteria, adaptions to the self-help booklet to make reference to a wider range of example activities, making reference to modifying goals, bringing discussion of "functional activities" forward, and providing a large print version of the self-help booklet when needed.

## Discussion

The BASIL trial is an external pilot trial designed to test acceptability of an adapted intervention and to refine trial procedures and design prior to undertaking a full-scale trial [22,39]. Our main finding is that higher-risk older people with LTCs living under COVID-19 restrictions were receptive to an approach to participate in a trial of a behavioural intervention. When offered BA, they preferred telephone contact rather than an offer of technology-enabled video calling. Levels of engagement with BA were high, with a greater proportion completing 6 or more planned sessions. Some people with LTCs declined the BASIL offer of telephone support.

In qualitative interviews, it was clear that those with very mild depression and good adaptation to socially-isolating restrictions were not an appropriate target group. This has led us to refine and target our intervention in a fully powered trial, and we will now only focus on older people who have some depressive symptoms above a threshold and at risk of further deterioration in mental health.

Although underpowered to test effectiveness, the between-group comparisons using CIs included benefit for BA in mitigating levels of depression at 3 months. For our measure of loneliness, there was good evidence of benefit and was unlikely to be a chance finding. Our preliminary analysis is in line with a CI approach to the interpretation of pilot trials [40], and we are keen not to overinterpret the positive finding of mitigating loneliness using BA. However, this is an encouraging finding that triangulates with the theoretical basis of BA in social isolation and the evidence of engagement and feasibility in the pilot trial. This justifies the need for a full-scale trial [41], where the consistency of this effect will be tested with greater levels of power and precision. The BASIL+ trial (the fully powered follow-on trial) is now underway and is preregistered (https://doi.org/10.1186/ISRCTN63034289) to reflect the design adaptations from the pilot study.

The BASIL trial and nested qualitative work adds to an emerging literature on the use of psychological interventions that incorporate cognitive or behavioural strategies to address loneliness and its causal role in depression [42]. Research to date has shown behavioural approaches to be highly effective in the treatment of depression among older people [18,20,43,44], and the preliminary results of the BASIL trial lend support to this approach in the face of COVID-19 restrictions and in mitigating loneliness [41]. A fully powered trial of

BA is now underway, and in time, this will report on the short- and long-term clinical and cost-effectiveness of a scalable behavioural psychosocial intervention. This will add to an emerging trial-based literature to establish the clinical and cost-effectiveness of interventions that target loneliness [12,45].

Our pilot trial was also undertaken rapidly and during the COVID-19 pandemic in early 2020. As such, we, along with other researchers undertaking trials during COVID-19, have had to adapt the methods used to generate randomised evidence. We have shown that it is possible to deliver trials with adaptations to minimise patient contact and streamline recruitment procedures. This makes us confident that this is an efficient method of participant engagement and follow-up for future trials, both under COVID-19 and beyond the pandemic [46]. It is of note that the time elapsed between the onset of the pandemic and the recruitment of the first participant was less than 3 months. We chose to study the impact of a plausible psychosocial intervention to mitigate depression and loneliness in an at-risk population of older people with multimorbidity. Population surveys under COVID-19 have shown that younger people are also at risk of loneliness [47] and psychological deterioration [48]. It is important that interventions to tackle the higher rates of depression and loneliness in all age groups are also developed and evaluated. Finally, we note that we worked with experts by experience at all stages of the design and delivery of the BASIL trial, and we believe the high levels of engagement reflect the positive contribution made by older people with lived experience in the BASIL programme.

The BASIL study had 2 main limitations. First, we found that the intervention was still being delivered at the prespecified primary outcome point, and this fed into the design of the main trial where a primary outcome of 3 months is now collected. Second, this was a pilot trial and was not designed to test between group differences with high levels of statistical power. Type 2 errors are likely to have occurred, and a larger trial is now underway to test for robust effects and replicate signals of effectiveness in important secondary outcomes such as loneliness. Finally, we acknowledge that we have relied on self-reported activity and social contact, and we ultimately do not know if this did in fact increase as a consequence of the intervention. The BASIL trial remains in follow-up, and we will report 12-month outcomes when these become available.

At the outset of the COVID-19 pandemic, it was predicted that there would be significant impacts on public mental health [5], including loneliness and depression, as a consequence of pandemic restrictions. This has come to pass [48], and population surveys indicate increased reports of loneliness and reports of depression [2]. The pandemic has also prompted a number of studies to understand the impacts of COVID-19 [49], but there have been very few studies to evaluate psychosocial interventions to mitigate psychological impact [46]. To our knowledge, BASIL is the first study to report trial-based evidence.

A clinical priority and policy imperative is to identify a brief and scalable intervention to prevent and mitigate loneliness, particularly in older people [50,51]. The preliminary results are in line with potential benefit for this intervention in mitigating loneliness at 3 months. We will now test the short- and long-term clinical and cost-effectiveness. This evidence may prove to be useful in improving the mental health of populations during the time of COVID-19 and also in mitigating depression and loneliness in socially isolated at-risk populations after the pandemic has passed [15].

## Supporting information

**S1 Table. Details of BASIL participants who did and did not complete the BASIL modules and included in qualitative analysis.** BASIL, Behavioural Activation in Social Isolation. (DOCX)

**S2 Table. Details of BSWs included in qualitative analysis. BSW, BASIL support worker.**
(DOCX)

**S1 Data. SAP.** SAP, Statistical Analysis Plan.
(PDF)

**S2 Data. CONSORT checklist.** CONSORT, Consolidated Standards of Reporting Trials.
(DOC)

## Acknowledgments

We would like to thank the participants for taking part in the trial, general practice and North East and North Cumbria Local Clinical Research Network staff for identifying and facilitating recruitment of participants, the independent Programme Steering Committee members for overseeing the study, and our PPI AG members for their insightful contributions and collaboration.

## Author Contributions

**Conceptualization:** Simon Gilbody, Elizabeth Littlewood, Dean McMillan, Carolyn A. Chew-Graham, Della Bailey, Samantha Gascoyne, Peter Coventry, Andrew Henry, Catherine Hewitt, Gemma Traviss-Turner, Karina Lovell, Sarah Dexter Smith, Judith Webster, David Ekers.

**Data curation:** Simon Gilbody, Elizabeth Littlewood, Dean McMillan, Samantha Gascoyne, Claire Sloan, Lauren Burke, Catherine Hewitt, Eloise Ryde, Rebecca Woodhouse.

**Formal analysis:** Elizabeth Littlewood, Claire Sloan, Caroline Fairhurst, Catherine Hewitt, Kalpita Joshi, Andrew Clegg, David Ekers.

**Funding acquisition:** Simon Gilbody, Elizabeth Littlewood, Carolyn A. Chew-Graham, Gemma Traviss-Turner, Andrew Clegg, Tom Gentry, Andrew J. Hill, Karina Lovell, David Ekers.

**Investigation:** Andrew Henry, Eloise Ryde, Leanne Shearsmith, Andrew Clegg.

**Methodology:** Elizabeth Littlewood, Carolyn A. Chew-Graham, Caroline Fairhurst, Catherine Hewitt, Kalpita Joshi, Eloise Ryde, Leanne Shearsmith, Gemma Traviss-Turner, Rebecca Woodhouse, Andrew Clegg, David Ekers.

**Project administration:** Simon Gilbody, Dean McMillan, Lauren Burke, Eloise Ryde, Leanne Shearsmith, Gemma Traviss-Turner, David Ekers.

**Resources:** Dean McMillan, Della Bailey, Claire Sloan, Peter Coventry, Andrew J. Hill, Judith Webster.

**Software:** Caroline Fairhurst.

**Supervision:** Simon Gilbody, Dean McMillan, Carolyn A. Chew-Graham, Della Bailey, Samantha Gascoyne, Suzanne Crosland, Rebecca Woodhouse, Tom Gentry, Andrew J. Hill, Sarah Dexter Smith.

**Validation:** Dean McMillan.

**Visualization:** Leanne Shearsmith.

**Writing – original draft:** Simon Gilbody, Dean McMillan, Carolyn A. Chew-Graham, Claire Sloan, Caroline Fairhurst, Catherine Hewitt, Kalpita Joshi, David Ekers.

**Writing – review & editing:** Simon Gilbody, Dean McMillan, Carolyn A. Chew-Graham, Della Bailey, Samantha Gascoyne, Claire Sloan, Lauren Burke, Peter Coventry, Suzanne Crosland, Caroline Fairhurst, Andrew Henry, Catherine Hewitt, Kalpita Joshi, Eloise Ryde, Leanne Shearsmith, Gemma Traviss-Turner, Rebecca Woodhouse, Andrew Clegg, Tom Gentry, Andrew J. Hill, Karina Lovell, Sarah Dexter Smith, Judith Webster, David Ekers.

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
