## [Editor Report · Decision Letter 0]

7 May 2021

Dear Dr Gilbody, 

Thank you for submitting your manuscript entitled "Mitigating the psychological impacts of COVID-19 restrictions: The Behavioural Activation in Social Isolation (BASIL) pilot randomised controlled trial to prevent depression and loneliness among older people with long term conditions" for consideration by PLOS Medicine.

Your manuscript has now been evaluated by the PLOS Medicine editorial staff as well as by an academic editor with relevant expertise and I am writing to let you know that we would like to send your submission out for external peer review.

Please re-submit your manuscript within two working days, i.e. by May 11 2021 11:59PM.

Kind regards,

Louise Gaynor-Brook, MBBS PhD

Senior Editor

PLOS Medicine

---

## [Decision Letter · Decision Letter 1]

23 Jun 2021

Dear Dr. Gilbody,

Thank you very much for submitting your manuscript "Mitigating the psychological impacts of COVID-19 restrictions: The Behavioural Activation in Social Isolation (BASIL) pilot randomised controlled trial to prevent depression and loneliness among older people with long term conditions" (PMEDICINE-D-21-02049R1) for consideration at PLOS Medicine. 

Your paper was sent to three independent reviewers, including a statistical reviewer, and has been discussed among all the editors here and with an academic editor with relevant expertise. The reviews are appended at the bottom of this email and any accompanying reviewer attachments can be seen via the link below:

[LINK]

In light of these reviews, I am afraid that we will not be able to accept the manuscript for publication in the journal in its current form, but we would like to consider a revised version that addresses the reviewers' and editors' comments. Obviously we cannot make any decision about publication until we have seen the revised manuscript and your response, and we plan to seek re-review by one or more of the reviewers. 

We expect to receive your revised manuscript by Jul 14 2021 11:59PM. Please email us (plosmedicine@plos.org) if you have any questions or concerns.

We look forward to receiving your revised manuscript. 

Sincerely,

Louise Gaynor-Brook, MBBS PhD

Associate Editor 

PLOS Medicine

plosmedicine.org

General comments:

Please provide line numbers (preferably not beginning from 1 on each new page) in your revised manuscript

Throughout the paper, please adapt reference call-outs to the following style: "... physical health and life expectancy [7,8]." (noting the absence of spaces within the square brackets, and space prior to the square brackets).

The terms gender and sex are not interchangeable; please use the appropriate term.

PLOS Medicine requires that the de-identified data underlying the specific results in a published article be made available, without restrictions on access, in a public repository or as Supporting Information at the time of article publication, provided it is legal and ethical to do so. Your Data Availability Statement (DAS) requires revision. If the data are owned by a third party but freely available upon request, please note this and state the owner of the data set and contact information for data requests (web or email address). Note that a study author cannot be the contact person for the data. If the data are not freely available, please describe briefly the ethical, legal, or contractual restriction that prevents you from sharing it. Please also include an appropriate contact (web or email address) for inquiries (which also cannot be a study author).

Title: Please revise your title according to PLOS Medicine's style. Please place the study design in the subtitle (ie, after a colon). We suggest “Behavioural activation to prevent depression and loneliness among socially isolated older people with long term conditions: the BASIL pilot randomised controlled trial” or similar 

Abstract:

Please report your abstract according to CONSORT for abstracts, following the PLOS Medicine abstract structure (Background, Methods and Findings, Conclusions), combining the Methods and Findings sections into one section http://www.consort-statement.org/extensions?ContentWidgetId=562

Abstract Background: Provide expand upon the context of why the study is important; e.g. were older adults becoming more socially isolated prior to the COVID-19 pandemic, or has this primarily arisen due to restrictions? The final sentence should clearly state the study question.

Abstract Methods and Findings:

Please provide brief demographic details of the study population (e.g. sex, age, ethnicity, etc)

Please specify the main outcome measures, and provide the number of participants lost to follow up in each group.

Please quantify the main results with 95% CIs and p values. Please include the important dependent variables that are adjusted for in the analyses.

Please be more specific in sentence ending “...but only a small number of participants had completed the intervention at this point”

In the last sentence of the Abstract Methods and Findings section, please describe 2-3 of the main limitation(s) of the study's methodology."

Please begin your Abstract Conclusions with "In this study, we observed ..." or similar, to summarize the main findings from your study.

Author Summary:

In the final bullet point of ‘What Do These Findings Mean?’, please describe the main limitation(s) of the study in non-technical language.

Introduction:

Please indicate whether your study is novel and how you determined that (being careful to temper assertions of primacy with ‘to the best of our knowledge’ or similar). 

Please remove the comma in the sentence beginning ‘Behavioural Activation (BA) is a practical treatment…’ 

Please remove reference to ‘ existing NIHR-funded programme of work’ (perhaps replacing with ‘built on upon our previous work to…’ or similar) and conclude the Introduction with a clear description of the study question or hypothesis, such as ‘In this pilot randomised controlled trial, we sought to assess…’, outlining your primary and secondary outcome measures.

Methods:

Please include the study protocol document and analysis plan, with any amendments, as Supporting Information to be published with the manuscript if accepted.

Please include a completed CONSORT checklist as Supporting Information. Please add the following statement, or similar, to the Methods: "This study is reported as per the Consolidated Standards of Reporting Trials (CONSORT) guideline (S1 Checklist)." The guideline can be found here: http://www.consort-statement.org/ When completing the checklist, please use section and paragraph numbers, rather than page numbers.

Please define BASIL at first use in the main text 

Please remove the sentence beginning ‘Here we report the preliminary results…’

Please remove website address https://www.nihr.ac.uk/covid-studies/studydetail.htm?entryId=249030 and cite this in your references

Please omit references to outcome measures and data collection at 12 months, as this is not reported/available. This could be added to your Discussion section as future work/next steps.

Please provide a reference to the figure (CONSORT flow diagram) provided.

Please add an ethics statement to your Methods section, providing the name(s) of the institutional review board(s) that provided ethical approval. Please specify whether informed consent was written or oral.

Results: 

The trial registration lists other secondary outcomes; can these be presented as part of this manuscript? If not, could you please indicate why this is not possible?

Generally the style should be "Database searches were conducted in two general practices…", for example. Numbers should be written out at the start of sentences, please. 

‘Other reasons’ for exclusion from randomisation should be defined.

Please do not use ‘average’ - please specify if this mean

No references to Table 2 is made in the main text of your Results section. 

PLOS does not permit ‘data not shown’. For the adjusted analyses presented in ’outcome data and between group comparisons at 1 and 3 months’ and Table 2, please also provide the unadjusted analyses and indicate which factors are adjusted for in the main text. 

Please remove the subheading ‘summary of findings’ 

“Intervention participant and BSW demographics are reported in the appendices. “ - please be more specific eg. Table S2

I realise that the patient quotes provided are written verbatim, but please consider adding extra detail in brackets to aid understanding, such as for “ I think the wide spread that you’ve currently can be...”

Please provide some more detail on the study adaptations - particularly eligibility criteria 

Discussion:

Please present and organize the Discussion as follows: a short, clear summary of the article's findings; what the study adds to existing research and where and why the results may differ from previous research; strengths and limitations of the study; implications and next steps for research, clinical practice, and/or public policy; one-paragraph conclusion.

Please remove all subheadings within your Discussion e.g. Main findings and other considerations

Please revise sentence beginning ‘This will add to an emerging…’ to omit ‘what works’ 

Tables:

Table 2 - where the mean score is presented for each of the scores used, please indicate the maximum score for context. Please define abbreviations used in the table legend.

Please provide the unadjusted comparisons as well as the adjusted comparisons in Table 3.

Please specify the variables adjusted for in the legend of Table 3.

References:

Please ensure that journal name abbreviations match those found in the National Center for Biotechnology Information (NCBI) databases, and are appropriately formatted and capitalised e.g. ref 8.

Please also see https://journals.plos.org/plosmedicine/s/submission-guidelines#loc-references for further details on reference formatting. 

Supplementary files: 

Please provide titles and legends for each individual table in the Supporting Information.

Please define abbreviations used in the table legend of each table. Please present numerators and denominators for percentages. 

Comments from the reviewers:

Reviewer #1: Thank you for the opportunity to review this pilot study, looking at behavioural activation in older adults during the COVID-19 pandemic. It is an important, interesting, and well written manuscript, and I have only a few comments below.

Page 5 - how were "older adults at risk loneliness and depression as a consequence of social isolation under COVID-19 restrictions" specifically defined? What were the risk factors?

Page 7 - I would like to see some more detail on the intervention itself. I'm particularly interested in examples of the "replacement" activities suggested to participants. 

Page 12 - There look to be quite considerable differences between the treatment and control groups at baseline, including differences in gender, LTC type, smoking status, education, and current circumstances. The authors do note the differences but state that there was a "reasonable balance in baseline characteristics". I am not sure I agree with this statement. Could the authors discuss this as a limitation further? 

There are also some differences between PHQ-9 at baseline, and the ES within groups at one month and comparable for both the treatment and control groups (around 0.2). There are some shifts in severity categories (table 2) that look different between groups, however it is not clear whether this is significant. Have the authors looked at reliable improvement or reliable improvement, which would allow us to see whether the movement above or below clinical thresholds is significant? If not for the pilot, then as part of the full-scale trial?

I am impressed by the high levels of engagement. Do the authors have any thoughts on the reasons for this or comparable studies that have (or have not) achieved such high completion rates?

Reviewer #2: Thank you for the opportunity to review this interesting paper describing an external pilot trial of a behavioural activation programme for older adults with long term health conditions. The paper is clearly written and the trial appears to have been conducted and reported in accordance with the ISRCTN-registered protocol. Changes to the trial aims and design from the originally funded programme, in response to the pandemic context, are clearly described. Learning from this pilot trial and resulting changes to the protocol for the future main trial are also clearly set out. 

Methods:

1. Process evaluation: report how many and which people were involved in data analysis, coding transcripts and reviewing/developing a coding frame. 

Results:

1. I think the authors should report statistical comparisons between groups for all of the secondary outcomes or none of them. Loneliness is only listed as one of several secondary outcomes in the ISRCTN protocol - after anxiety. Just reporting one secondary outcome risks looking like the authors have cherry-picked the most positive result. The descriptive data for GAD-7 looks a lot less promising - was this result significant? To avoid any chance of misinterpretation, I think the authors should also avoid saying that non-significant results "favoured" the intervention group (p15 and throughout). 

2. I think the abstract should be correspondingly amended - just report the feasibility outcomes and primary outcome, or all secondary outcomes too.

3. The appendices reporting the characteristics of the participants and support workers should be more clearly labelled to make it clear that these are the characteristics of those participating in the qualitative interviews

The accompanying qualitative interviews are only reported very briefly, but this may be due to space constraints, and it's good to see a separate publication will follow which can do more justice to this work. It is helpful to have even the brief summary of findings provided here. 

4. The brief summary of study adaptations on p.17 could be helpfully described more fully - in an appendix if space is short in the main paper. 

Discussion:

Overall I thought the discussion was clear, fair and to the point. I think the authors are right to conclude that this pilot demonstrates clearly that a trial of this intervention is feasible and acceptable, and there is an exciting signal of efficacy for loneliness (less clear for depression - see below). I think two limitations should be briefly acknowledged and one conclusion qualified. 

1. The authors should acknowledge as a limitation that they haven't measured, or even apparently got an indication from the qualitative interviews, of whether the intervention did increase people's activity or social contact with others. The process evaluation hasn't been able to determine whether the intervention achieved the intended behaviour change which may lead to reduced depression or loneliness.

2. The authors should also acknowledge more explicitly that the pilot study took place in the highly unusual covid pandemic context, when many older adults with long term conditions were shielding or diligently socially-distancing. It can't be certain that recruitment and retention would be as easy or outcomes would be similar in a changing covid or post-covid context. (Although I agree with the authors that their recruitment and retention rates are so good, it bodes well for non-covid contexts too.)

3. I think the concluding statement on p.19 is overstated " The preliminary results are in line with potential benefit for this intervention at 3 months". For the primary depression outcome, mean PHQ-9 scores for the treatment group dropped by less than a point during the study; the three month between-groups comparison offers no indication of efficacy; and even at the top end of the confidence intervals, the between group difference is less than two points, i.e. less than the threshold on PHQ-9 for when patients are likely to notice a meaningful change (Kounali et al. 2020). This should be briefly discussed and the conclusion qualified accordingly. I agree the results are more promising for loneliness. 

And finally, it's good news that a full trial of the revised programme is already underway.

4. Please provide the ISRCTN reference number for the full trial and a reference for the weblink so this can be easily found by interested readers. 

Kounali D et al. (2020). How much change is enough? Evidence from a longitudinal study on depression in UK primary care. Psychological Medicine 1-8. https://doi.org/10.1017/S0033291720003700

Reviewer #3: Thanks for the opportunity to review your manuscript. My role is as a statistical reviewer, so my questions and comments focus on the study design, data, and analysis presented. I have put general comments and questions first and followed these with queries relevant to a specific part of the manuscript (with a page and paragraph number reference).

This manuscript presents 'BASIL', a pilot RCT that tests an intervention of behavioural activation against usual care in older adults (>65) with chronic health conditions from general practices in the UK. The behavioural activation component was delivered remotely with the COVID-19 restrictions in mind. The primary aim of the pilot was assessment of the feasibility and acceptability of the intervention. Investigators also collected PHQ9 and other secondary measures including HR-QOL, anxiety, and loneliness. 

94% of participants were in 1-month follow-up, and 90% at three months of follow-up, with a high proportion of patients recruited after screening. The qualitative component indicated good acceptability of the BA as well. All together indicating a promising piloted intervention. This is an impressive effort as it was organised and run during the first wave of COVID-19 (and to think I was just moping around at home in my trackpants during the same time-period). I have some queries that I would like resolved but this manuscript should be considered again after revision.

How was the missing data accounted for in the analyses? It looks as though a complete case analysis was used, which I think is fine considering efficacy isn't the main consideration, but it should be clear that this approach was used and what the limitations are of assuming MCAR. 

Was the study conducted across all the UK or focused in a particular region?

P3, Paragraph 3. Could you add the number of available sessions to the info about median session completed, e.g. (median 6 sessions out of 8 available)?

Paragarph 4. 'unlikey to be due to chance' seems like a clumsy way to describe this - maybe 'good evidence' is better? 

P4, Paragraph 1. Would it be better to qualify 'social isolation' as 'physical social isolation'?

P6, Paragraph 2. Could the complete list of physical conditions be included somewhere in the manuscript (even as a supp appendix)? There is list of some in Table 1 but there are quite a few people with 'other'. 

 P7, Paragraph 4. What was the typical type of care that risky patients were referred to? How many were referred to more formal healthcare interventions during the study? 

P9, Paragraph 1. Is the SAP available to be reviewed with the manuscript? Also, could the full protocol be attached, I tried to follow the reference number in the methods but got the 'page not found' each time I tried to load it (user error hopefully).

Clustering by BSW was included in the sample size calculations, was this also accounted for in the main analyses?

How was baseline adjusted for? By using a change-score in the analysis, or by including the baseline score as a covariate ('ANCOVA' approach)? 

Were the key assumptions of the general linear model checked, e.g. distribution of residuals, linearity of baseline score (assuming the ANCOVA approach was used), no highly influential observations? 

P12, Table 1. Could you add a footnote clarifying the differences between 'Shielding' and the other distancing/isolating categories?

P12, Paragraph 1. Yes, a bit of imbalance but probably about what you would expect for that number of baseline characteristics and nothing I'd be too worried about. 

P15, Paragraph 1. Is the unadjusted mean difference presented in the results? The easiest way to add this would probably be to add a column to the Table 3.

This is the first time statistical significance has been mentioned - what level of alpha was used? Or was this based on the confidence limits? It might be better to just discuss the CI for scores and talk about the relative level of evidence for efficacy e.g. no evidence, some evidence, good evidence etc. (not that I would expect to see this for a pilot study). 

P18. Paragraph 1. My own view on pilot studies (there is a good discussion about this in Westlund and Stuart 2018: https://doi.org/10.1177/1098214016651489) is that the theoretical grounds for why an intervention may work, along with the feasibility and acceptability are what really drive the justification for a full evaluation. The efficacy findings are promising but I don't think it's these that justify a full study.

Figure 1. It would be easier to track the withdrawls and incomplete data if these weren't cumulative - I couldn't figure out whether patients are discontinued if they did not complete a questionnaire at month 1, e.g. in the usual care arm the number of patients goes from 49 to 45 (loss of 4), but there is 1 full withdrawal and 4 incomplete questions. Did one of the patients fully withdraw and also not complete questionnaire? 

Supp Appendix

Table 1. It would be easier to read the table with the columns for completers and non-completers next to each other. Some more detailed table titles would be helpful - I assume 'non-completers' are participants in the intervention arm who didn't complete the modules, not study non-completers.

[LINK]

---

## [Decision Letter · Decision Letter 2]

11 Aug 2021

Dear Dr. Gilbody,

Thank you very much for re-submitting your manuscript "Behavioural activation to prevent depression and loneliness among socially isolated older people with long term conditions: the BASIL COVID-19 pilot randomised controlled trial" (PMEDICINE-D-21-02049R2) for review by PLOS Medicine.

I have discussed the paper with my colleagues and the academic editor and it was also seen again by three reviewers. I am pleased to say that provided the remaining editorial and production issues are dealt with we are planning to accept the paper for publication in the journal.

[LINK]

We look forward to receiving the revised manuscript by Aug 18 2021 11:59PM.   

Sincerely,

Louise Gaynor-Brook, MBBS PhD

Associate Editor 

PLOS Medicine

plosmedicine.org

Requests from Editors:

General comments:

Please adapt reference call-outs to the following style: "... as depression in older people [4,6]." (noting the absence of spaces within the square brackets).

Please see https://journals.plos.org/plosmedicine/s/supporting-information for our supporting information guidelines.

Thank you for updating your data availability statement - please ensure that this is added to the metadata accompanying your manuscript. 

I note that “ Requests for data sharing will be considered by SG and the independent trial steering and data monitoring committee”; in general, authors should not be involved in decisions relating to data access so please confirm that data requests will be assessed primarily by the independent committee and please revise your statement accordingly.

Thank you for providing an Author Summary. Please organise this into lists of 2-3 bullet points per question. Please review the text provided for the Author Summary carefully as grammatical errors are present e.g. “They are risk of social isolation”. Please also adhere to the standard PLOS headings - ‘What Do These Findings Mean?’. In the final bullet point of ‘What Do These Findings Mean?’, please describe the main limitation(s) of the study in non-technical language, which might include that this pilot trial was not designed to test differences in outcomes between the two groups.

Introduction:

Line 161 - please consider revising to ‘brief effective intervention’

Methods:

Please mention your statistical analysis plan early in the Methods section and refer to the relevant supplementary file. 

Line 227 - Please specify whether informed consent was written or oral.

Thank you for providing the CONSORT checklist. When completing the checklist, please use section and paragraph numbers, rather than page or line numbers which will likely no longer correspond to the appropriate sections after copy-editing.

Line 313 - Please refer to the relevant supplementary file name for the CONSORT checklist. 

Results: 

Line 449 - please refer to individual file names - e.g. Table S1 rather than an appendix (Please see https://journals.plos.org/plosmedicine/s/supporting-information). 

Discussion:

Line 503 - please consider revising ‘very low’ to ‘very mild’

In the paragraph beginning “Our pilot trial was also undertaken…” please consider revising ‘during COVID’ and ‘under COVID’ to ‘during the COVID-19 pandemic’ or similar 

Line 553 - please consider revising to ‘At the outset of the COVID-19 pandemic…’

In Table 1, supplementary tables 1 and 2 - please replace 'gender' with 'sex' (including footnote for supplementary table 2)

References:

Please ensure that journal name abbreviations match those found in the National Center for Biotechnology Information (NCBI) databases, and are appropriately formatted and capitalised e.g. ref 47

Supplementary files

Please replace 'gender' with 'sex' in your SAP file 

Comments from Reviewers:

Reviewer #1: The authors have addressed my previus comments. I have no further comments. 

Reviewer #2: The authors have adequately addressed my previous comments except in these two respects:

Results comment 1: The authors said they have removed the term "favoured" where this was used to describe non-significant differences between groups. But they haven't in lines 430 and 432: I still think they should.

Results comment 2: I recommended that the abstract should just report the primary outcomes and the feasibility outcomes; or all the secondary outcomes, not just selected secondary outcomes. The authors said they have amended the abstract to incorporate this suggestion. They have now listed all the secondary outcomes in the abstract methods, but in the abstract results, they still report the results for one secondary outcome (loneliness) but not the other secondary outcomes. I think they should amend this as I suggested, and report results for all secondary outcomes or none in the abstract. 

If the authors address these two points to the editors' satisfaction, I think this paper is very suitable for publication in PLOS Medicine without need for further review. 

Reviewer #3: Thanks for the revised manuscript and responses to my original queries.

I have reviewed the SAP - this matches the analysis described in the manuscript, thanks for including this.

The changes and responses all address my original review well and I have no further requested revisions.

[LINK]

---

## [Editor Report · Decision Letter 3]

20 Aug 2021

Dear Prof. Gilbody, 

On behalf of my colleagues and the Academic Editor, Prof. Vikram Patel, I am pleased to inform you that we have agreed to publish your manuscript "Behavioural activation to prevent depression and loneliness among socially isolated older people with long term conditions: the BASIL COVID-19 pilot randomised controlled trial" (PMEDICINE-D-21-02049R3) in PLOS Medicine.

PRESS

Sincerely, 

Louise Gaynor-Brook, MBBS PhD 

Associate Editor 

PLOS Medicine